

# iTRAQ-based quantitative proteome analysis reveals metabolic changes between a cleistogamous wheat mutant and its wild-type wheat counterpart

Caiguo Tang[1,2,*], Huilan Zhang[1,2,*], Pingping Zhang[3], Yuhan Ma[1], Minghui Cao[1,2], Hao Hu[1,2], Faheem Afzal Shah[1], Weiwei Zhao[1], Minghao Li[1,2] and Lifang Wu[1]

[1] Key laboratory of High Magnetic Field and Ion beam physical biology, Hefei Institutes of Physical Science, Chinese Academy of Sciences, Hefei, Anhui, China
[2] School of Life Sciences, University of Science and Technology of China, Hefei, Anhui, China
[3] School of Life Sciences, Anhui University, Hefei, Anhui, China
* These authors contributed equally to this work.

Corresponding authors
Minghao Li, limh@ipp.ac.cn
Lifang Wu, lfwu@ipp.ac.cn

## ABSTRACT

**Background:** Wheat is one of the most important staple crops worldwide. *Fusarium* head blight (FHB) severely affects wheat yield and quality. A novel bread wheat mutant, *ZK001*, characterized as cleistogamic was isolated from a non-cleistogamous variety Yumai 18 (YM18) through static magnetic field mutagenesis. Cleistogamy is a promising strategy for controlling FHB. However, little is known about the mechanism of cleistogamy in wheat.
**Methods:** We performed a FHB resistance test to identify the FHB infection rate of *ZK001*. We also measured the agronomic traits of *ZK001* and the starch and total soluble sugar contents of lodicules in YM18 and *ZK001*. Finally, we performed comparative studies at the proteome level between YM18 and *ZK001* based on the proteomic technique of isobaric tags for relative and absolute quantification.
**Results:** The infection rate of *ZK001* was lower than that of its wild-type and Aikang 58. The abnormal lodicules of *ZK001* lost the ability to push the lemma and palea apart during the flowering stage. Proteome analysis showed that the main differentially abundant proteins (DAPs) were related to carbohydrate metabolism, protein transport, and calcium ion binding. These DAPs may work together to regulate cellular homeostasis, osmotic pressure and the development of lodicules. This hypothesis is supported by the analysis of starch, soluble sugar content in the lodicules as well as the results of Quantitative reverse transcription polymerase chain reaction.
**Conclusions:** Proteomic analysis has provided comprehensive information that should be useful for further research on the lodicule development mechanism in wheat. The *ZK001* mutant is optimal for studying flower development in wheat and could be very important for FHB resistant projects via conventional crossing.

## INTRODUCTION

Bread wheat (*Triticum aestivum* L.) is one of the most important staple crops worldwide. The world population continues to grow and arable area is decreasing year by year, therefore higher production in crop plants may prove to be necessary to satisfy the increasing demand for food (*FAO, 2015*). However, many challenges, including biotic and abiotic stresses, severely affect wheat fields, and product quality. For instance, *Fusarium* head blight (FHB) is critically damaging wheat security (*Walter, Nicholson & Doohan, 2010*), and the application of chemical insecticides and fungicides is increasing the amounts of residues in wheat and in the environment (*Hollingsworth et al., 2008*). Therefore, scientists and breeders have to find eco-friendly and cost-effective strategies to guarantee wheat yield and quality.

In the last decade, severe epidemics caused by *Fusarium* spp. have occurred worldwide with up to 100% yield loss recorded under optimal disease conditions (*Yumurtaci et al., 2017*). The *pore-forming toxin-like* gene at the quantitative trait locus *Fhb1,* which was the first FHB-resistance gene isolated, was found to confer resistance to FHB in Sumai 3 (SM3) (*Rawat et al., 2016*). FHB infection usually occurs on the inner surfaces of lemmae and paleae after germination of the *Fusarium* spp. conidia (*Zange, Kang & Buchenauer, 2005*). The anther can provide the initial path for FHB infection (*Pugh, Johann & Dickson, 1933*; *Walter, Nicholson & Doohan, 2010*); therefore, cleistogamous cultivars, which contain few anthers exposed to glumes, may provide structural barriers for diseases that appear during the flowering stage. In barley (*Hordeum vulgare* L.), cleistogamous cultivars, which self-fertilize within permanently closed flowers (*Culley & Klooster, 2007*), showed greater resistance to FHB infection than chasmogamous cultivars, which have open flowers (*Yoshida, Kawada & Tohnooka, 2005*). In wheat, cleistogamous cultivars such as U24 have a lower risk of FHB infection than chasmogamous cultivars such as Saikai 165 (*Kubo et al., 2010*). Therefore, cleistogamy might be the basis for new strategies for controlling FHB in cereal crops.

Cleistogamy in barley is genetically determined by the presence of the recessive allele *cly1*, but the dominant allele at the linked locus, *Cly2*, is epistatic over *cly1* (*Wang et al., 2013*). Loss of the miRNA172 target site causes *cly1* to express a protein, *Hv*AP2, which effectively suppresses lodicule swelling (*Turuspekov et al., 2004*; *Nair et al., 2010*; *Wang et al., 2015*). In rice (*Oryza sativa* L.), there are many cleistogamous mutants resulting from abnormal lodicules. A single recessive gene, *lodiculeless spikelet(t)* [*ld(t)*], controls the cleistogamous mutant lacking lodicules (*Won & Koh, 1998*; *Maeng et al., 2006*). Another rice mutant, which has a truncated DEP2 determined by the *cl7(t)* gene, has a cleistogamous phenotype because of weak swelling ability in the lodicules (*Ni et al., 2014*). A third rice mutant, *spw1-cls*, has normal stamens, but the lodicules are transformed homeotically into lodicule-glume mosaic organs, thereby engendering cleistogamy with temperature-sensitivity (*Yoshida et al., 2007*; *Ohmori et al., 2012*). A novel temperature-stable cleistogamous mutant, *spw1-cls2*, can maintain the cleistogamous phenotype under low temperatures (*Lombardo et al., 2017*). The glumes open in the flowering stage because the swelling of the lodicule is primarily responsible for pushing the

lemma and palea, thereby opening the floret (*Nair et al., 2010*). In contrast, there is very little information on cleistogamy in wheat.

The probability of primary infection is approximately proportional to the number of spores reaching the open florets during the flowering process; accordingly, the breeding of varieties with flowers that are partially or completely cleistogamous might reduce *Fusarium* susceptibility in wheat (*Schuster & Ellner, 2008*). In order to probe the mechanism of cleistogamy, *Ning et al. (2013)* studied the structure, transcription, and post-transcriptional regulation of the cleistogamous gene *TaAP2*, which is homologous in barley and wheat. *TaAP2* alleles may also generate a cleistogamous wheat and improve resistance to FHB. Additionally, anther extrusion is a complex trait with significant markers; it has either favorable or unfavorable additive effects and imparts minor to moderate levels of phenotypic variance in spring and winter wheat (*Muqaddasi et al., 2017*).

Large-scale transcriptomic analyses have been employed in wheat to better understand the molecular mechanisms of flower development (*Winfield et al., 2010*; *Diallo et al., 2014*; *Feng et al., 2015*; *Kumar et al., 2015*; *Yang et al., 2015*; *Ma et al., 2017*). However, because of post-transcriptional and post-translational modifications, mRNA levels do not always correlate with the corresponding protein levels (*Schweppe et al., 2003*; *Canovas et al., 2004*; *Zhao et al., 2013*). Proteins are directly correlated with cellular functions (*Yan et al., 2005*; *Zhang et al., 2012*); therefore, proteomic analysis is essential for studying global protein expression levels in wheat to further unravel the complex mechanisms of cleistogamy. In particular, isobaric tags for relative and absolute quantitation (iTRAQ) technology, which is a quantitative gel-free proteomic approach, coupled with liquid chromatography–tandem mass spectrometry (LC-MS/MS) enables the direct quantification and comparison of protein levels among samples with more efficiency and accuracy than traditional gel-based techniques which fail to identify low-abundance protein species and have limitations for identifying proteins with extreme biochemical properties (*Wu et al., 2006*).

Here, we show that an SMFs-induced wheat mutant, *ZK001*, is a cleistogamous line with lower FHB infection vulnerability than the chasmogamous line Yumai 18 (YM18). Additionally, we performed a comparative proteomic analysis of different development stages in YM18 and *ZK001* to characterize the protein expression profiles. In this manner, we aimed to provide insight into proteomic changes associated with the cleistogamous phenotype in wheat, specifically exploring lodicule expanding mechanisms at the protein level. Our results have the potential to benefit future research efforts in controlling FHB via conventional breeding, advance the study of wheat flower development, and contribute to better control of genetically modified lines of agriculturally important crops; this could lead to time and cost-savings in the effort to refine genotypes.

# MATERIALS AND METHODS

## Plant material

The cleistogamy mutant line *ZK001* was isolated from a mutagenized population of wheat variety YM18, using an SMFs of 7 Tesla for 5 h. After mutagenesis, it propagated via self-pollination until the cleistogamous phenotype was completely stable. All wheat seeds
were stored at the Hefei Institutes of Physical Science, Chinese Academy of Sciences (CASHIPS), Anhui, P. R. China. SM3, Aikang 58 (AK58), YM18, and *ZK001* were grown in a greenhouse in an experimental field (31°54′N, 117°10′E) at CASHIPS. AK58 and YM18, which are both varieties in the northern China, were not resistant to FHB infection (*Yu et al., 2019*). Therefore, AK58 was used as susceptible control. Fertilizer and weed management were similar to methods used in the process for wheat breeding (*Li et al., 2014*). The spikelets and lodicules of YM18 and *ZK001*, which had three biological replicates, were harvested during the white anther stage (WAS), green anther stage (GAS), yellow anther stage (YAS), and anthesis stage (AS) (*Zadoks, Chang & Konzak, 1974*; *Kirby & Appleyard, 1987*; *Guo & Schnurbusch, 2015*). These samples were collected and frozen in liquid nitrogen and preserved at −80 °C.

## Starch and total soluble sugar content

A total of 20 pairs of lodicules with three biological replications from YM18 and *ZK001* were sampled and snap frozen in liquid nitrogen at the four flower development stages. The samples were ground using Tissuelyser-24 (Shanghai Jingxin Industrial Development Co., Ltd., Shanghai, China) for 45 s at 50 Hz.

Starch and total soluble sugars were extracted following the instructions included with the Starch Content and Plant Soluble Sugar Content test kits (Nanjing Jiancheng Bioengineering Institute, Nanjing, China). The starch and total soluble sugars in the supernatant were determined using a UV–VIS spectrophotometer (Lambda 365; PerkinElmer, Waltham, MA, USA) with a wavelength of 620 nm. The starch and total soluble sugar contents were calculated using the following formulas:

$$C_{\text{starch}} = \frac{\text{OD}_{\text{sample}} - \text{OD}_{\text{blank}}}{\text{OD}_{\text{standard}} - \text{OD}_{\text{blank}}} \times \frac{C_{\text{standard 1}} \times V_{\text{pretreatment}} \times \text{Dilution ratio}}{N_{\text{total PL}}}$$

$$C_{\text{total souble sugar}} = \frac{\text{OD}_{\text{sample}} - \text{OD}_{\text{blank}}}{\text{OD}_{\text{standard}} - \text{OD}_{\text{blank}}} \times \frac{C_{\text{standard 2}} \times \text{Dilution ratio}}{10 \times V_{\text{distrilled water}} \times N_{\text{total PL}}}$$

Note: PL, pair of lodicules; $C_{\text{starch}}$, the starch content in lodicule (µg·PL$^{-1}$); $C_{\text{total souble sugar}}$, the total souble sugar content in lodicule (µg·PL$^{-1}$); $C_{\text{standard 1}}$, standard solution 1 concentration = 200 µg·mL$^{-1}$; $C_{\text{standard 2}}$, standard solution 2 concentration = 100 µg·mL$^{-1}$, Dilution ratio = 1; $V_{\text{pretreatment}}$, the volume of pretreatment solution = 1.7 mL; $V_{\text{distilled water}}$, the volume of distilled water used for homogenising = one mL; $N_{\text{total PL}}$, the total number of PL which were sampled = 20.

## Observation of spikes and lodicules

Spike images of YM18 and *ZK001* were photographed (D90; Nikon, Tokyo Metropolis, Japan) at the AS. After 90 min, the lodicules of YM18 and *ZK001*, which were separated from the central young spikes in triplicate during GAS and cultured on Murashige and Skoog (MS) medium, were observed with an upright fluorescence stereomicroscope (SZX10; Olympus, Tokyo, Japan) and photographed (DP72; Olympus, Tokyo, Japan).

## FHB resistance testing

*Fusarium* head blight resistance testing was performed during the flowering stage of SM3, AK58, YM18, and *ZK001* in the greenhouse by spraying the FHB spore F0601

(*Fusarium graminearum* Schw. cv. F0601) in both 2013–2014 and 2014–2015. The inoculum (50 µL at 105 spores per mL) was deposited by spraying both sides of the ears. The diseased spikelet rate was calculated using the following formula:

$$\text{Diseased spikelets rate} = \frac{N_{\text{infected spikelets}}}{N_{\text{total spikelets}}} \times 100\%.$$

Note: $N_{\text{infected spikelets}}$, the number of infected spikelets; $N_{\text{total spikelets}}$, the number of total spikelets.

## Protein extraction and iTRAQ labelling

Total soluble proteins were extracted according to a published procedure (*Yang et al., 2013*) with slight modifications. Briefly, moderate amounts of the samples were separately frozen using liquid $N_2$ and ground in −20 °C pre-cooled pestles and mortars with urea extraction buffer containing 150 mM Tris–HCl (pH 7.6), 8M urea, 0.5% SDS, 1.2% Triton X-100, 20 mM EDTA, 20 mM EGTA, 50 mM NaF, 1% glycerol 2-phosphate, five mM DTT, and 0.5% phosphatase inhibitor mixture 2 (Sigma, Darmstadt, Germany). The mixtures were centrifuged at $10,000 \times g$ for 1 h at 4 °C, then the supernatants were mixed with pre-cooled acetone/methanol and incubated for 1 h at −20 °C. The mixtures were centrifuged at $15,000 \times g$ for 15 min at 4 °C. The pellets were washed twice with cold acetone. Pellets were dried and solubilized in lysis buffer containing 50 mM Tris–HCl (pH 6.8), 8M urea, five mM DTT, 1% SDS, and 10 mM EDTA. Protein concentrations of the samples were estimated using the Bradford method (*Bradford, 1976*) (Table S1) and the samples were stored at −80 °C for further use. All protein samples were checked via sodium dodecyl sulfate polyacrylamide gel electrophoresis (SDS–PAGE) according to the Schägger protocol (*Schägger, 2006*). SDS-PAGE gels (Fig. S1) were stained with Coomassie Brilliant Blue staining solution (Coomassie Blue Fast Staining Solution, Beijing Dingguo Changsheng Biotechnology Co., Ltd., Beijing, China) (*Kang et al., 2002*).

After determining the protein concentration, we digested the samples with trypsin (V5113; Promega, Madison, WI, USA) and then incubated them for 12–16 h at 37 °C. Approximately 100 µg of peptides of the different samples were labelled with iTRAQ based on the protocol of *Unwin, Griffiths & Whetton (2010)*. The peptides of the different samples were labelled with iTRAQ reagents (isobaric tags 113, 114, 115, 116, 117, 118, 119, and 121 for groups YM18-WAS, *ZK001*-WAS, YM18-GAS, *ZK001*-GAS, YM18-YAS, *ZK001*-YAS, YM18-AS, and *ZK001*-AS, respectively) according to the manufacturer's instructions (Applied Biosystems, Foster City, CA, USA).

## HPLC grading of C$_{18}$ columns at high pH and LC-electrospray ionization-MS/MS analysis

The lyophilized peptide mixture was reconstituted with 100 µL of solution A (2% acetonitrile (ACN) and 20 mM ammonium formate, pH 10). Then, the samples were loaded onto a reverse-phase column (C$_{18}$ column, 1.9 µm (particle size), 150 µm (inner diameter) × 120 mm (length), Waters) and eluted using a step linear elution program (Table S2). The samples were collected every 1.5 min and centrifuged at $14,000 \times g$ for 5–90 min. The 60 collected fractions were dried and re-dissolved with five µL 0.5% formic

acid (FA). The collected fractions were finally combined into 10 pools and centrifuged at $14,000 \times g$ for 10 min.

The reconstituted peptides were analyzed with a Q-Exactive HF mass spectrometer (Thermo Fisher Scientific, Waltham, MA, USA) coupled with a nano high-performance liquid chromatography system (1260 Infinity II; Agilent, Santa Clara, CA, USA) (*Scheltema et al., 2014*). The peptides were loaded onto a $C_{18}$ reversed-phase column ($C_{18}$ column, three μm (particle size), 100 μm (inner diameter) $\times$ 200 mm (length), Thermo Scientific, Waltham, MA, USA) using mobile phases A (0.1% $FA/H_2O$) and B (0.08% FA, 80% ACN) (Table S3). The HPLC effluent was directly electrosprayed into the mass spectrometer and analyzed based on pre-set parameters (Fig. S2).

## Data analysis

The raw mass data were processed for peptide identification using Proteome Discoverer 1.4 (ver. 1.4.0.288; Thermo Fisher Scientific, Waltham, MA, USA) with specific parameters (Table S4) for searching the UniProt *Triticum* database. A false discovery rate (FDR) of $\leq 0.01$ was estimated for protein identification using a target-decoy search strategy (*Elias & Gygi, 2007*). The mass spectrometry proteomics data have been deposited in the ProteomeXchange Consortium (http://proteomecentral.proteomexchange.org) via the PRIDE partner repository (*Vizcaino et al., 2016*) with the dataset identifier < PXD010188 >. Increasing and decreasing abundant proteins were determined based on 1.5-fold-changes and peptides spectral matches (PSMs) $\geq 2$ (*Sharma et al., 2017*) between *ZK001*-WAS and YM18-WAS (Group 1), *ZK001*-GAS and YM18-GAS (Group 2), *ZK001*-YAS and YM18-YAS (Group 3), *ZK001*-AS and YM18-AS (Group 4), YM18-WAS and YM18-GAS (Group 5), YM18-WAS and YM18-YAS (Group 6), YM18-WAS and YM18-AS (Group 7), *ZK001*-WAS and *ZK001*-GAS (Group 8), *ZK001*-WAS and *ZK001*-YAS (Group 9), and *ZK001*-WAS and *ZK001*-AS (Group 10).

Protein annotation was conducted by a BLAST search against NCBI and UniProt databases. Protein function was classified based on the following databases: Gene Ontology (http://www.geneontology.org/, GO), and Kyoto Encyclopedia of Genes and Genomes (http://www.genome.jp/kegg/, KEGG). For analysis of differentially abundant proteins (DAPs), significant GO enrichment and KEGG enrichment were defined as a corrected FDR with a *P*-value less than 0.01 (*Benjamini & Hochberg, 1995*). Proteins containing at least two PSMs per protein and fold change ratios $\geq 1.5$ or $\leq 0.67$ were considered more abundant or less abundant proteins, respectively. In order to validate the DAPs profile, we searched the EnsemblPlants database (http://plants.ensembl.org/index.html) for corresponding DNA sequences. A total of 10 DAPs involved in carbohydrate metabolism and calcium ion binding and transport were selected for Quantitative reverse transcription polymerase chain reaction (qRT-PCR) validation.

## Quantitative real-time PCR validation

Total RNA was isolated using a Plant RNA kit (R6827; Omega, Norcross, America) according to the manufacturer's instructions. The quality of each RNA sample was checked on 1% agarose gels. Measurement of the concentration of RNA samples was

**Table 1 Comparison of diseased spikelets rate by *Fusarium* head blight in 2013–2014 and 2014–2015 in four varieties.**

| Variety | Diseased spikelets rate[a] (%) | |
| --- | --- | --- |
| | 2013–2014 | 2014–2015 |
| SM3 | 7.03 ± 2.23 | 8.61 ± 3.40 |
| *ZK001* | 9.39 ± 3.31 | 17.60 ± 3.60 |
| YM18 | 15.20 ± 2.46 | 35.16 ± 3.71 |
| AK58 | 20.41 ± 6.76 | 38.12 ± 6.13 |

Note:
[a] Diseased spikelets rate = the number of infected spikelets/total number of spikelets × 100%. The results presented are the means of three independent experiments. Error bars, s.d.

performed using a NanoDrop 2000 spectrophotometer bioanalyzer (Thermo Fisher Scientific, Waltham, MA, USA). cDNAs were synthesized using TransScript One-Step gDNA Removal and cDNA Synthesis SuperMix (Transgen Biotech, Beijing, China) according to the manufacturer's protocol. qRT-PCR was used to measure the transcript levels of the proteins of interest. Each experiment was performed in three technical replicates with three biological replicates. Target gene-specific primers (Table S5) were designed using the online software Primer 3 version 0.4.0 (http://bioinfo.ut.ee/primer3-0.4.0/primer3/) (*Untergasser et al., 2012*). qRT-PCR was performed according to the manufacturer's instructions for the FastStart Essential DNA Green Master (Roche, Basel, Switzerland), run on the Roche LightCycler® 96 Instrument. The *glyceraldehyde-3-phosphate dehydrogenase* gene from *T. aestivum* (*TaGAPDH*, GI: 7579063) served as an internal control and the relative expression of target genes was calculated using the $2^{-\Delta\Delta CT}$ method (*Livak & Schmittgen, 2001*).

## Statistical data analysis

The experimental data values represented the average of the measurements conducted from three independent assays and were expressed as the mean ± standard error of the mean. The data were further analyzed using ANOVA followed by Duncan's test (SPSS 18.0; IBM, Somers, NY, USA). The level of significance was set at $P \leq 0.05$.

# RESULTS

## Comparative resistance to *Fusarium* head blight

The results of FHB resistance testing showed that the infection rate in SM3 and *ZK001* were 7.03% and 9.39% in 2013–2014 and 8.61% and 17.60% in 2014–2015, respectively (Table 1). Compared to SM3 and *ZK001*, YM18, and AK58 were highly susceptible: the FHB infection rate was 35.16% and 38.12% in 2014–2015, respectively. However, the diseased spikelet rates for YM18 and AK58 were 15.20% and 20.41% in 2013–2014, respectively, which was half the rate in 2014–2015 (Table 1). This indicates that the FHB infection rate is greatly influenced by environmental factors. These results suggest that cleistogamous cultivars have a lower FHB infection rate than chasmogamous cultivars.

## Comparison of flowering in YM18 and *ZK001*

In accordance with previous reports, the exserted anthers increased the incidence of FHB (*Sage & De Isturiz, 1974*; *Parry, Jenkinson & McLeod, 1995*). Furthermore, the anthers of

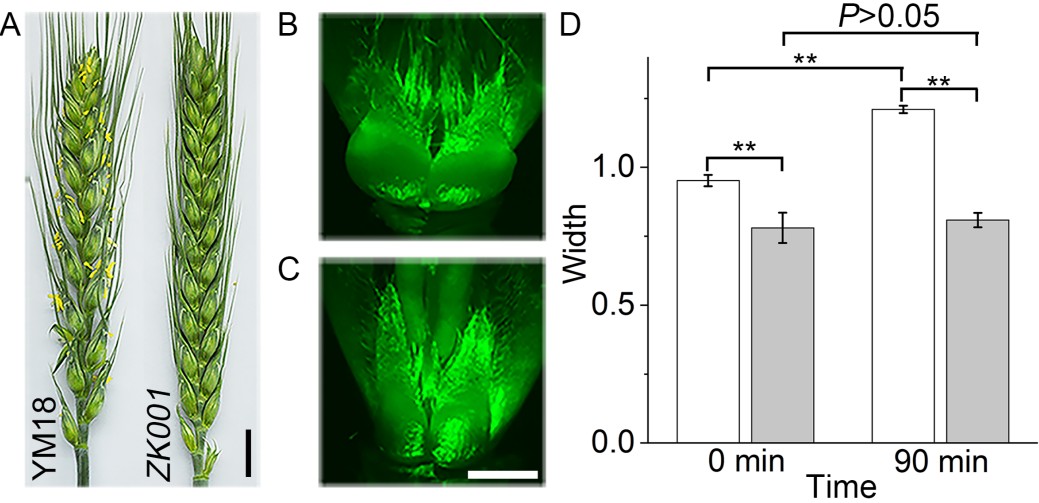

**Figure 1 Characteristics of genotypes in the two differential individual lines of YM18 and *ZK001*.** Comparison of the inflorescence details between YM18 and *ZK001* in post-anthesis stage (Bar = one cm) (A). Lodicules of YM18 (B) and *ZK001* (C) which were sampled in GAS and cultured on MS media containing graphite were observed after 90 min by microscope (Bar = one mm). (D) Comparison of lodicule width between YM18 (white column) and *ZK001* (gray column). The results presented are the means of four independent experiments expressed as the mean ± standard error of the mean (SEM). The data were further analyzed using an ANOVA at a 95% confidence level following Duncan's test (SPSS 18.0, IBM, Somers, NY, USA). The level of significance was set at $P \leq 0.05$ (*) or $P \leq 0.001$ (**).

cleistogamy wheat were detained in glums during the flowering stage. Although the individual lines of YM18 and *ZK001* were grown under the same growth and environmental conditions, the morphological differences were obvious. In YM18, the anthers extruded from the palea and lemma at the AS, whereas, no anthers were observed in *ZK001* at all flower development stages (Fig. 1A). The morphology of the lodicules of YM18 and *ZK001* was also obviously different. In order to investigate their morphology, we harvested the lodicules of YM18 and *ZK001* at the GAS and cultured them for 90 min on MS medium. The width of the YM18 lodicules (Fig. 1B) was greater than that of the *ZK001* lodicules (Figs. 1C and 1D).

## Physiological characteristics of lodicules in YM18 and *ZK001*

To reveal the cause of the lodicules difference between YM18 and *ZK001*, we measured the starch and total soluble sugar contents in the lodicules of YM18 and *ZK001* at the four flower development stages. Lodicule starch (Fig. 2A; Table S6) and total soluble sugar (Fig. 2B; Table S6) contents showed an overall increase from the WAS to the AS for YM18 and *ZK001*. No significant differences in the starch and soluble sugar contents in the lodicule were observed between YM18 and *ZK001* during the WAS or GAS. Additionally, the starch and total soluble sugar contents in *ZK001* during the YAS significantly decreased 2.40- and 1.75-fold, respectively (both $P < 0.05$), compared with those in YM18, detected in one pair of lodicules (Fig. 2; Table S6). In contrast, the starch and total soluble sugar contents in *ZK001* during the AS remarkably increased 3.57- and 1.52-fold, respectively (both $P < 0.05$), compared with those in YM18, detected in one pair of lodicules (Fig. 2; Table S6).

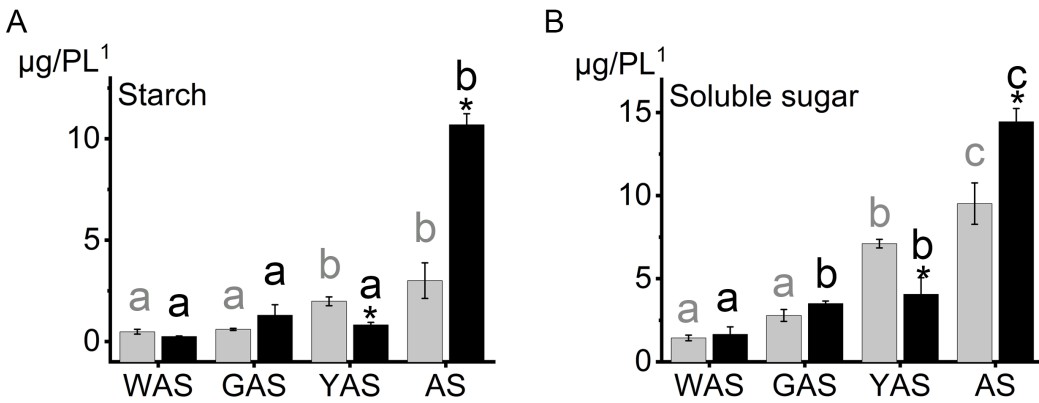

**Figure 2 Starch, soluble sugar content in lodicules of YM18 (gray column) and *ZK001* (black column).** (A) Comparison of starch content change tendency in lodicules between YM18 and *ZK001*. (B) Comparison of soluble sugar content change tendency in lodicules between YM18 and *ZK001*. PL[1]: pair of lodicules (PL). The results presented are the means of three independent experiments. Error bars, s.d. Columns marked with different lowercase letter indicate difference in means using the one-way ANOVA LSD analysis of PASW Statistics software among four flower development stage of YM18 (gray lowercase) and *ZK001* (black lowercase). The asterisk indicates the difference between YM18 and *ZK001* at WAS, GAS, YAS, and AS, respectively. The data were further analyzed using an ANOVA at a 95% confidence level following Duncan's test (SPSS 18.0; IBM, Somers, NY, USA). The level of significance was set at $P \leq 0.05$ or $P \leq 0.001$.

## Overview of quantitative proteome analysis

In order to study the protein expression patterns in YM18 and *ZK001*, we examined and quantitatively catalogued the proteomes of YM18 and *ZK001* in the four flower development stages using iTRAQ technology. In this study, 19,422 peptides were matched to 4,497 proteins in the samples (Table S7); in addition, 11,603 unique peptides were found, and 2,172 proteins were identified with more than two unique peptide sequences excluding post-translational modifications. As shown in Fig. 3A, more than 99% of the peptides covered proteins within the 36 peptides, and protein quantity decreased as the number of matching peptides increased. In terms of protein mass distribution, good coverage (an average of 10–18% of total proteins in each protein-mass group) was obtained for proteins >10 kDa and <60 kDa (Fig. 3B). The length of the identified peptides was between 10 and 13 amino acids at the peak and approximately 93% of the peptide length was within 24 amino acids (Fig. 3C). Over 77% of the proteins had >5% sequence coverage. Additionally, sequence coverage distribution was high in most of the identified proteins: More than 58% had over >10% coverage and more than 37% had over 20% coverage (Fig. 3D). These results indicate that the identified peptides were sufficient for protein identification.

## Cluster analysis of protein expression at four developmental stages

In order to identify more DAPs, we compared DAPs in YM18 and *ZK001* in the flowering development process, *ZK001*-WAS vs YM18-WAS (Group 1), *ZK001*-GAS vs YM18-GAS (Group 2), *ZK001*-YAS vs YM18-YAS (Group 3), *ZK001*-AS vs YM18-AS (Group 4), YM18-WAS vs YM18-GAS (Group 5), YM18-WAS vs YM18-YAS (Group 6), YM18-WAS vs YM18-AS (Group 7), *ZK001*-WAS vs *ZK001*-GAS (Group 8), *ZK001*-WAS vs *ZK001*-YAS (Group 9), and *ZK001*-WAS vs *ZK001*-AS (Group 10). Increasing abundance

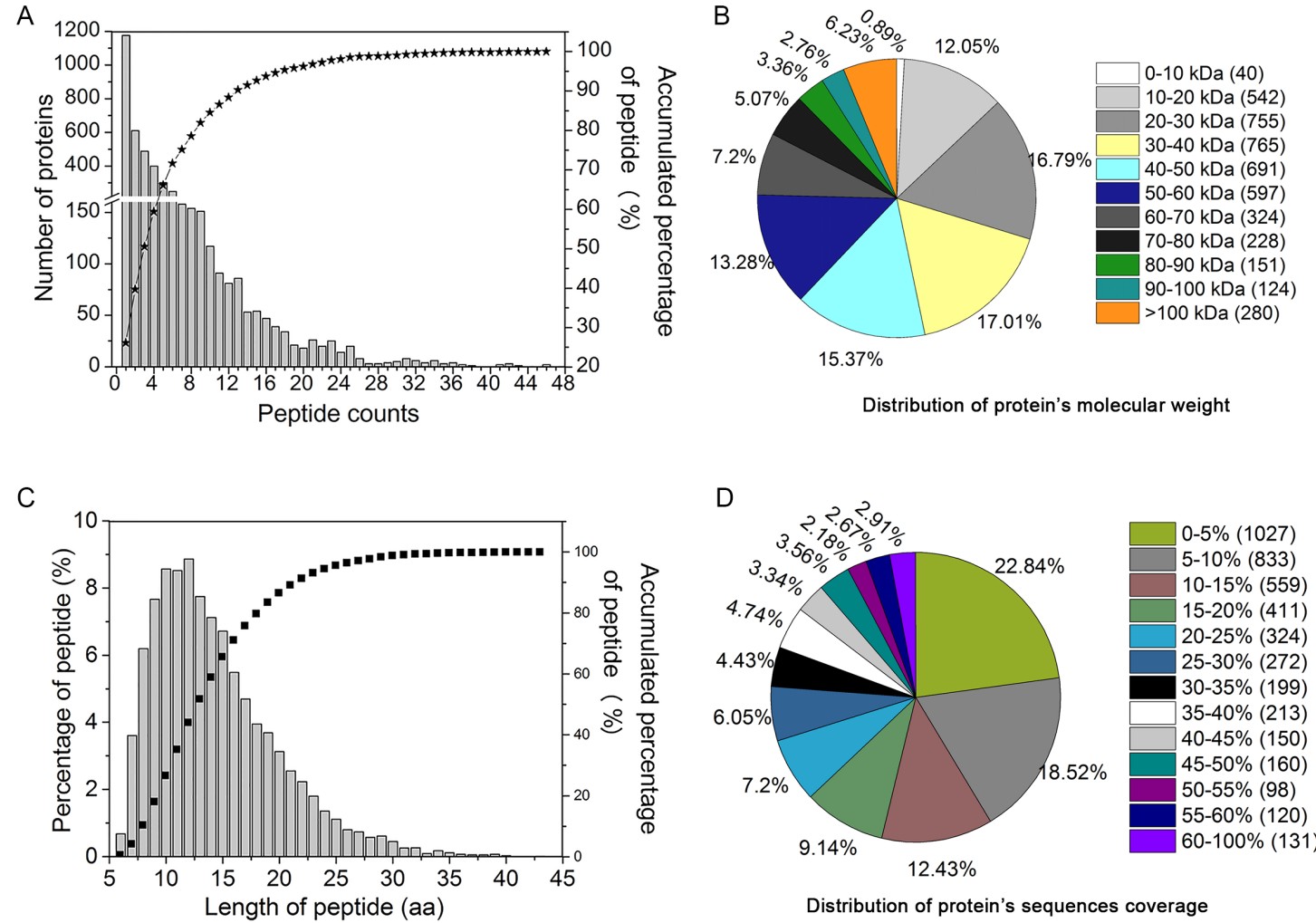

**Figure 3 Assessment of iTRAQ analysis for peptides identification and quantitation.** (A) The distribution of the identification peptide segments counts corresponding to the identification of proteins number. (B) Distribution of protein's molecular weight. (C) Quantification of peptide-length coverage in the identified proteins. (D) Coverage of protein mass distribution.

and decreasing abundance proteins were determined based on fold-changes (FC) of >1.5 or <0.667 for expression difference comparison. For further screening, approximately 16, 47, 2, 0, 11, 124, 105, 15, 298, and 188 DAPs were identified with a corrected $P$-value for GO KEGG enrichment less than 0.01 in groups 1 to 10 (Table 2). A Venn diagram of the DAPs and their overlap in Group 1 and Group 2 showed that two common DAPs were increased-abundance and one common DAP was decreased-abundance (Fig. S3). Group 3 and Group 4 showed no overlap with Group 1 or Group 2 (Fig. S3). Venn diagrams indicated that 6, 10, 32, 206, 57, and 140 DAPs were specific DAPs of Groups 5, 8, 6, 9, 7, and 10, respectively (Fig. S3).

## Functional classification and subcellular localization of proteins

Gene ontology analysis showed that all of the identified proteins in YM18 and *ZK001* were involved in 11 subgroups of MF, 19 subgroups of BP, and 14 subgroups of CC

**Table 2 The number of differentially abundant proteins (DAPs) at four flower development stages.**

| Groups | Total | Corrected *P*-value < 0.01 | | |
| --- | --- | --- | --- | --- |
| | | Increasing-DAPs | Decreasing-DAPs | Total DAPs |
| Group 1 | 4,188 | 7 | 9 | 16 |
| Group 2 | 4,188 | 9 | 38 | 47 |
| Group 3 | 4,189 | 1 | 1 | 2 |
| Group 4 | 4,189 | 0 | 0 | 0 |
| Group 5 | 4,188 | 0 | 11 | 11 |
| Group 6 | 4,188 | 27 | 97 | 124 |
| Group 7 | 4,188 | 31 | 74 | 105 |
| Group 8 | 4,188 | 4 | 11 | 15 |
| Group 9 | 4,188 | 123 | 175 | 298 |
| Group 10 | 4,188 | 94 | 94 | 188 |

Notes:
Group 1: *ZK001*-WAS vs YM18-WAS, Group 2: *ZK001*-GAS vs YM18-GAS, Group 3: *ZK001*-YAS vs YM18-YAS, Group 4: *ZK001*-AS vs YM18-AS, Group 5: YM18-WAS vs YM18-GAS, Group 6: YM18-WAS vs YM18-YAS, Group 7: YM18-WAS vs YM18-AS, Group 8: *ZK001*-WAS vs *ZK001*-GAS, Group 9: *ZK001*-WAS vs *ZK001*-YAS, and Group 10: *ZK001*-WAS vs *ZK001*-AS.
$0.667 < FC < 1.5$, corrected *P*-value < 0.01, PSMs $\geq$ 2.

(Fig. S4; Table S7). Significant GO enrichment was employed to analyze the DAPs with a corrected FDR *P*-value less than 0.01 and an FC ratio of more than 1.5. Based on GO annotations and enrichments, the DAPs of Group 1 were enriched in molecular function terms for lipid binding (100%) (Fig. 4A) as well as biological process terms for lipid transport (14.06%), lipid localization (14.06%), macromolecule localization (20.31%), organic substance transport (20.31%), single-organism transport (15.63%), and single-organism localization (15.63%) (Fig. 4B). GO classification of Group 2 revealed that the DAPs were enriched in the biological process, cellular component, and molecular function (Fig. 4C). No protein was enriched in Group 3 or Group 4. The DAPs of Groups 5 to 10 were also classified into biological process (Fig. S5A), cellular component (Fig. S5B) and molecular function (Fig. S5C). DAPs involved in carbohydrate metabolism and transport, calcium ion binding and protein transport, and fatty acid biosynthesis were further used in cluster analyses.

## Accumulation patterns of DAPs and verification of DAPs of interest

Based on the above analyses, 11 genes which corresponded to DAPs of interest were chosen for qRT-PCR analyses using gene-specific primers (Table S5) to explore the expression profile at the transcription level.

The lodicule morphology showed significant differences between YM18 and *ZK001* (Fig. 1). We performed qRT-PCR using the RNA of YM18 and *ZK001* lodicules to study the transcript profiles of the 11 genes corresponding to the selected DAPs (Fig. 5). The results of qRT-PCR indicated that the expression level of the gene *A0A1D6CCI3* (encoding the bidirectional sugar transporter SWEET) was expressed in the lodicules at the WAS and GAS of both YM18 and *ZK001*, with almost no expression at the YAS and AS (Fig. 3; Table S7); (Fig. 5A). Additionally, the expression level of the gene *A0A1D5WGA3* (encoding a nutrient reservoir-related protein) was extremely down-regulated from the

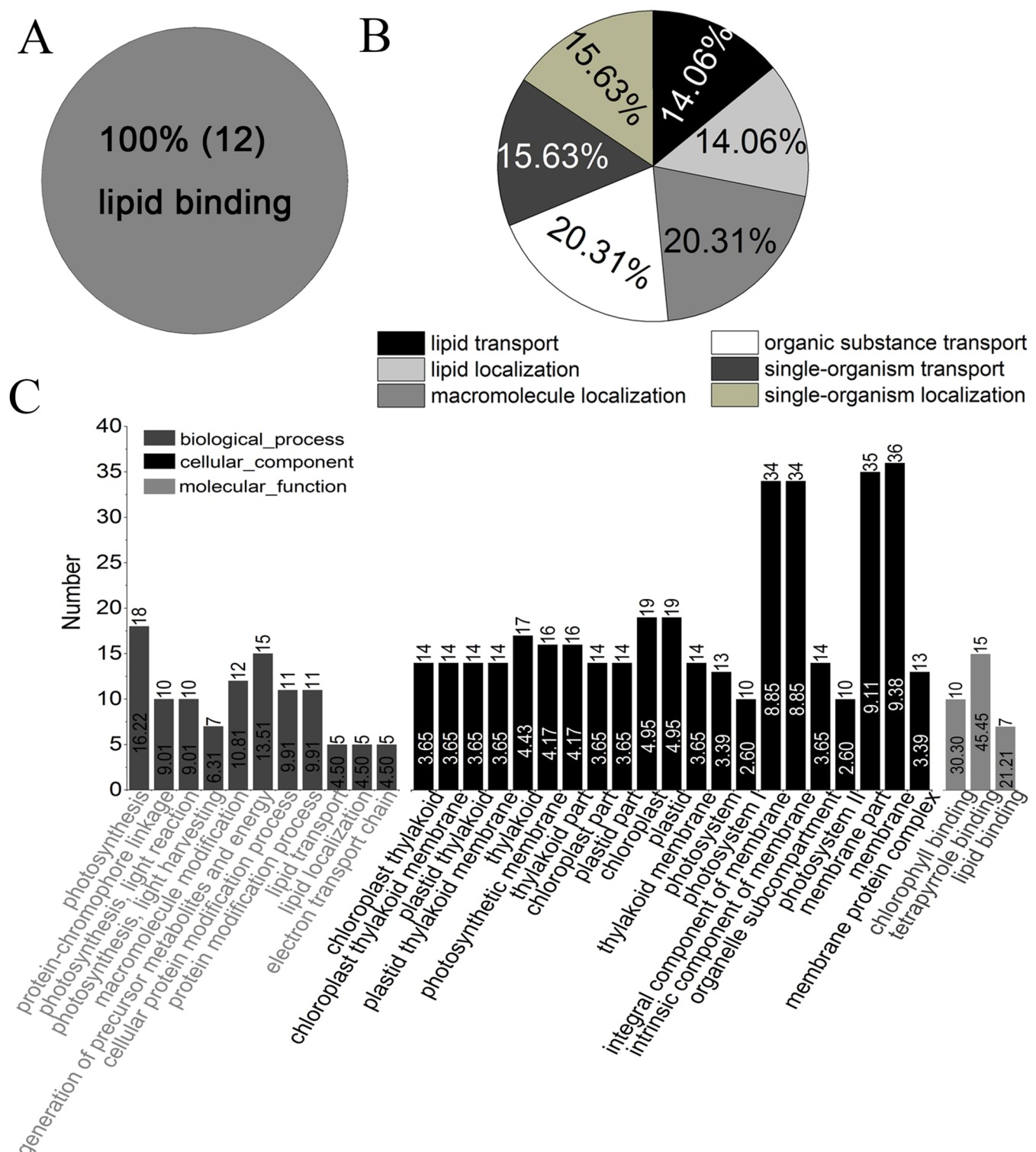

**Figure 4 GO classification of DAPs of Group 1 to 4 from four flower development stages in YM18 and *ZK001* based on GO enrichment.** (A) Molecular function of Group 1; (B) biological process of Group 1; (C) biological process, cellular component, and molecular function of Group 2. No protein was enriched in Group 3 or Group 4 basing on GO enrichment. ($0.667 <$ FC $< 1.5$, corrected *P*-value $< 0.01$, PSMs $\geq 2$).

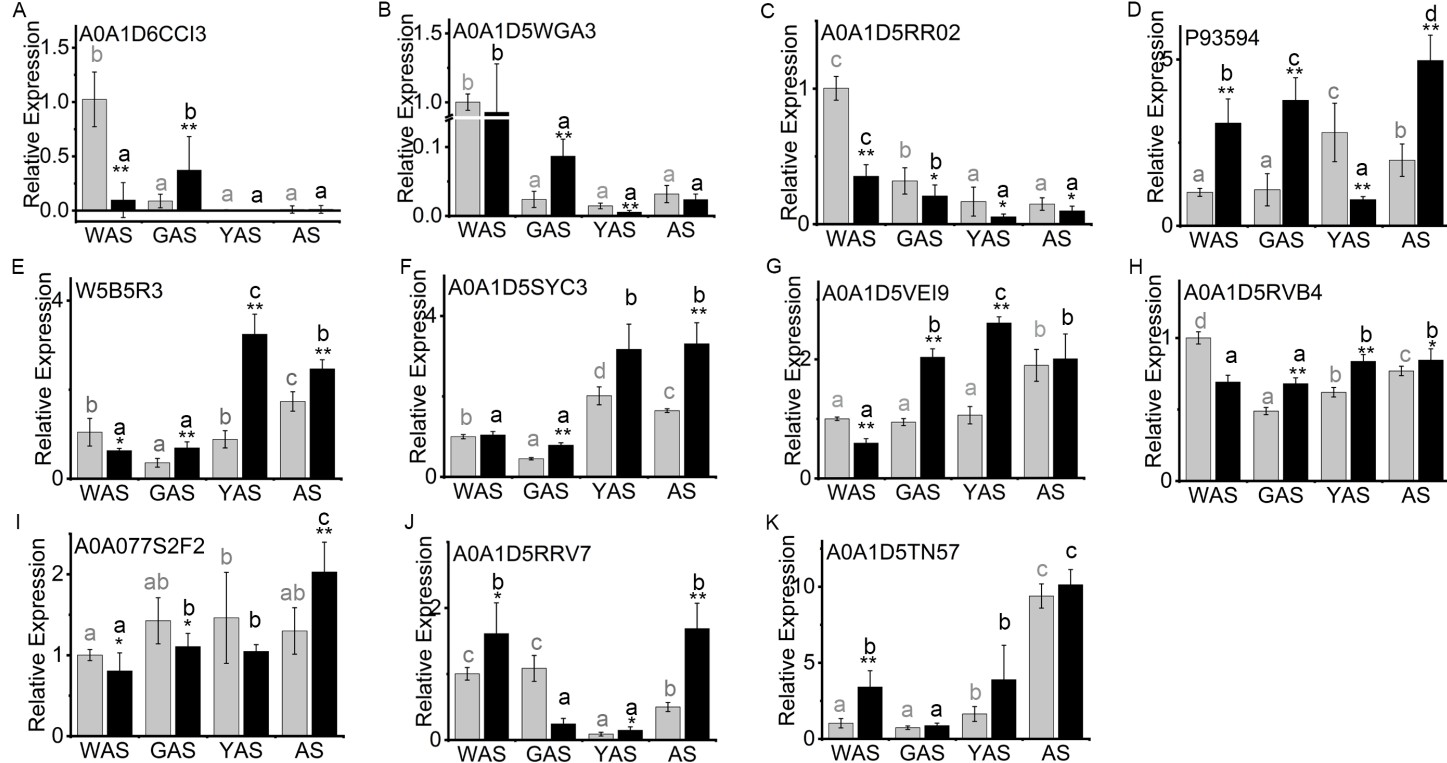

**Figure 5 Relative expression profile of related genes which corresponded to DAPs in lodicules of YM18 (gray column) and *ZK001* (black column).** (A–I) Relative expression profile of carbohydrate-related genes in lodicules of YM18 and *ZK001*. (J–K) Relative expression profile of calcium ion binding-related genes in lodicules of YM18 and *ZK001*. The results presented are the means of three independent experiments. Error bars, s.d. Columns marked with different lowercase letter indicate difference in means of YM18 (gray lowercase) and *ZK001* (black lowercase) using the one-way ANOVA LSD analysis of PASW Statistics software. The asterisk indicates the difference between YM18 and *ZK001* at WAS, GAS, YAS, and AS, respectively. The data were further analyzed using an ANOVA at a 95% confidence level following Duncan's test (SPSS 18.0; IBM, Somers, NY, USA). The level of significance was set at $P \leq 0.05$ or $P \leq 0.001$.

WAS to the GAS in both YM18 and *ZK001* (Fig. 5B). Though the relative expression level in *ZK001* was significantly higher than that in YM18 in the GAS ($P < 0.001$), the relative expression levels were all less than 0.1 in the GAS, YAS or AS in YM18 or *ZK001* (Fig. 5B). Compared with the relative expression levels of the gene encoding beta-amylase A0A1D5RR02 in YM18, the gene in *ZK001* was down-regulated during the WAS ($P < 0.001$), GAS ($P < 0.05$), YAS ($P < 0.05$), and AS ($P < 0.05$) (Fig. 5C). The relative expression levels of *A0A1D5RR02* were all down-regulated from the WAS to the YAS in YM18 and *ZK001*, but there was no significant difference from the YAS to the AS in YM18 or *ZK001* (both $P > 0.05$) (Fig. 5C). Additionally, compared with its expression in the lodicules of YM18, P93594 (beta-amylase)-encoding mRNA was all up-regulated ($P < 0.001$) in the lodicules of *ZK001*, except that in the YAS (Fig. 5D). Interestingly, the mRNA level of sucrose synthase W5B5R3 in *ZK001* indicated up-regulation compared with the levels in YM18 at the GAS, YAS, and AS ($P < 0.001$), and down-regulation at the WAS ($P < 0.05$) (Fig. 5E). The mRNA levels of *A0A1D5SYC3* (a gene encoding cellular glucose homeostasis-related proteins) and *A0A1D5VEI9* (a gene encoding cellular glucose

homeostasis-related proteins) showed almost the same expression profile as *W5B5R3*, except for that of *A0A1D5SYC3* in the WAS and that of *A0A1D5VEI9* in the AS (Figs. 5E–5G). The relative expression level of *A0A1D5RVB4* (a gene encoding cellular glucose homeostasis-related proteins) and *W5B5R3* were similar (Figs. 5E and 5H). Compared with expression levels in YM18, the gene expression of beta-glucosidase activity-related protein A0A077S2F2 in *ZK001* was down-regulated ($P < 0.05$) during the WAS and GAS and up-regulated ($P < 0.001$) in the AS (Fig. 5I).

Calcium ions play a key role in the development of plants. Therefore, the relative expression of the genes encoding calcium ion binding-related protein A0A1D5TN57 and annexin A0A1D5RRV7 were also evaluated to determine the profile during flower development. Compared with levels in YM18, the relative gene expression levels of A0A1D5TN57 in *ZK001* were up-regulated during the WAS and YAS (Fig. 5K), and those of A0A1D5RRV7 were up-regulated during the WAS, YAS, and AS (Fig. 5J). However, the expression levels of *A0A1D5RRV7* and *A0A1D5TN57* were all down-regulated ($P < 0.05$) from the WAS to the GAS in *ZK001* (Figs. 5J and 5K).

## DISCUSSION

### Cleistogamy provides structural barriers for diseases of *Fusarium* head blight

From the physiological point of view, the flowering stage is regarded as the most susceptible period for primary infection of wheat spikes by FHB because of the opening of wheat florets and the extension of anthers (*Pugh, Johann & Dickson, 1933*; *Schroeder & Christensen, 1963*; *Gilsinger et al., 2005*; *Schuster & Ellner, 2008*). Barley is a plant that self-fertilizes with permanently closed flowers, but chasmogamous barley varieties are easily infected with *Fusarium* (*Yoshida, Kawada & Tohnooka, 2005*; *Culley & Klooster, 2007*). Table 1 shows that the diseased spikelet rate in 2014–2015 was more severe than that in 2013–2014 except in SM3, possibly because of the resistance gene *Fhb1* (*Rawat et al., 2016*). Compared with YM18, which is a wild-type chasmogamous cultivar, the diseased spikelet rate for *ZK001*, a mutant cleistogamous cultivar, decreased by 38.22% and 50.00% in 2013–2014 and 2014–2015, respectively (Table 1). This indicates that although the diseased spikelet rate is greatly influenced by environmental factors, cleistogamous cultivars that flower partially or completely may have a lower risk of FHB infection than chasmogamous cultivars (*Kubo et al., 2010*; *Wang et al., 2013*). Therefore, we further verified the hypothesis that cleistogamous wheat cultivars might have lower *Fusarium* susceptibility. A practical strategy for controlling FHB would be to introduce the cleistogamous character into other varieties that are suitable for production and promotion but sensitive to FHB through hybridization.

### Lodicules play an important role in glume opening/closing in wheat

The molecular mechanism for cleistogamy has been intensively studied in rice (*Maeng et al., 2006*; *Yoshida et al., 2007*; *Ohmori et al., 2012*; *Ni et al., 2014*; *Lombardo et al., 2017*) and barley (*Turuspekov et al., 2004*; *Hori et al., 2005*; *Nair et al., 2010*; *Wang et al., 2013*, *2015*; *Zhang et al., 2016*). However, the molecular mechanism for cleistogamy in wheat

remains unclear, though it is known that the lodicule is a key factor in glume opening/closing in the monocotyledon. The abnormal lodicules may lack the ability to push the lemma and palea apart during the flowering stage in the cleistogamous mutant *ZK001* (Fig. 1). This phenomenon is similar to that occurring in barley (*Nair et al., 2010*).

## Carbohydrates and calcium are the main factors regulating lodicule osmotic pressure

Sucrose is the primary form of sugar transported for photosynthetic carbon assimilation (*Chen et al., 2012*). The *A0A1D6CCI3* gene was expressed in the lodicules of both YM18 and *ZK001* (Fig. 5A; Table S8). This indicates that carbohydrates can be transferred normally to the lodicules of both YM18 and *ZK001*. Nevertheless, the starch (Fig. 2A; Table S6) and soluble sugar (Fig. 2B; Table S6) contents of the YM18 lodicules increased dramatically from the GAS to the YAS. *Liu et al. (2017)* suggested that retarded lodicule expansion in *ZS97A* was caused by reduced water accumulation because of diminished accumulation of osmotic regulation substances. In contrast, the lower soluble sugar content in the lodicules of *ZK001* prevented the accumulation of water during the YAS (Fig. 2B; Table S6). The lodicule size of YM18 was larger than that of *ZK001* because the starch and soluble sugar contents in the lodicules of *ZK001* decreased from the GAS to the YAS, leading to little water transfer to the lodicules. The lodicules of wheat swell extensively and subsequently contract after rapid autolysis of the tissues (*Craig & O'Brien, 1975*). Accordingly, the starch and soluble sugar contents in the lodicule of YM18 were lower than those in *ZK001* at the AS (Figs. 2A and 2B; Table S6).

Cytosolic calcium is an important secondary messenger in plants and plays important roles in the response to both environmental and internal signals (*Poovaiah & Reddy, 1993*; *Liao, Zheng & Guo, 2017*). Plant annexins are calcium-dependent phospholipid binding proteins with many biological functions; for instance, they participate in calcium ion channel formation, membrane dynamics, plant growth, and the stress response (*Mortimer et al., 2008*; *Laohavisit & Davies, 2011*). In this study, the relative gene expression levels of annexin (A0A1D5RRV7) and the calcium ion binding-related protein (A0A1D5TN57) in the lodicules of *ZK001* were up-regulated during the WAS compared to those of in the lodicules of YM18 (Figs. 5J and 5K). Therefore, we infer that the WAS is a critical period for the lodicules.

## An overview of the pathways for proteome metabolic changes in lodicules

Many substances regulate the osmotic pressure of lodicules, such as soluble sugar (*Zee & O'Brien, 1971*; *Wang, Gu & Gao, 1991*; *Yan et al., 2017*), starch (*Pissarek, 1971*), calcium (*Qin, Yang & Zhao, 2005*; *Chen et al., 2016*), and potassium (*Heslop-Harrison & Heslop-Harrison, 1996*; *Chen et al., 2016*; *Liu et al., 2017*). Our findings, together with those of previous studies, provide an overview of the metabolic pathways involving the carbohydrates that regulate the osmotic pressure of the lodicules. As shown in Fig. 6, sucrose is transferred into the lodicules from the extra-cellular environment through a bidirectional sugar transporter (A0A1D6CCI3), and is converted into D-fructose-6P by

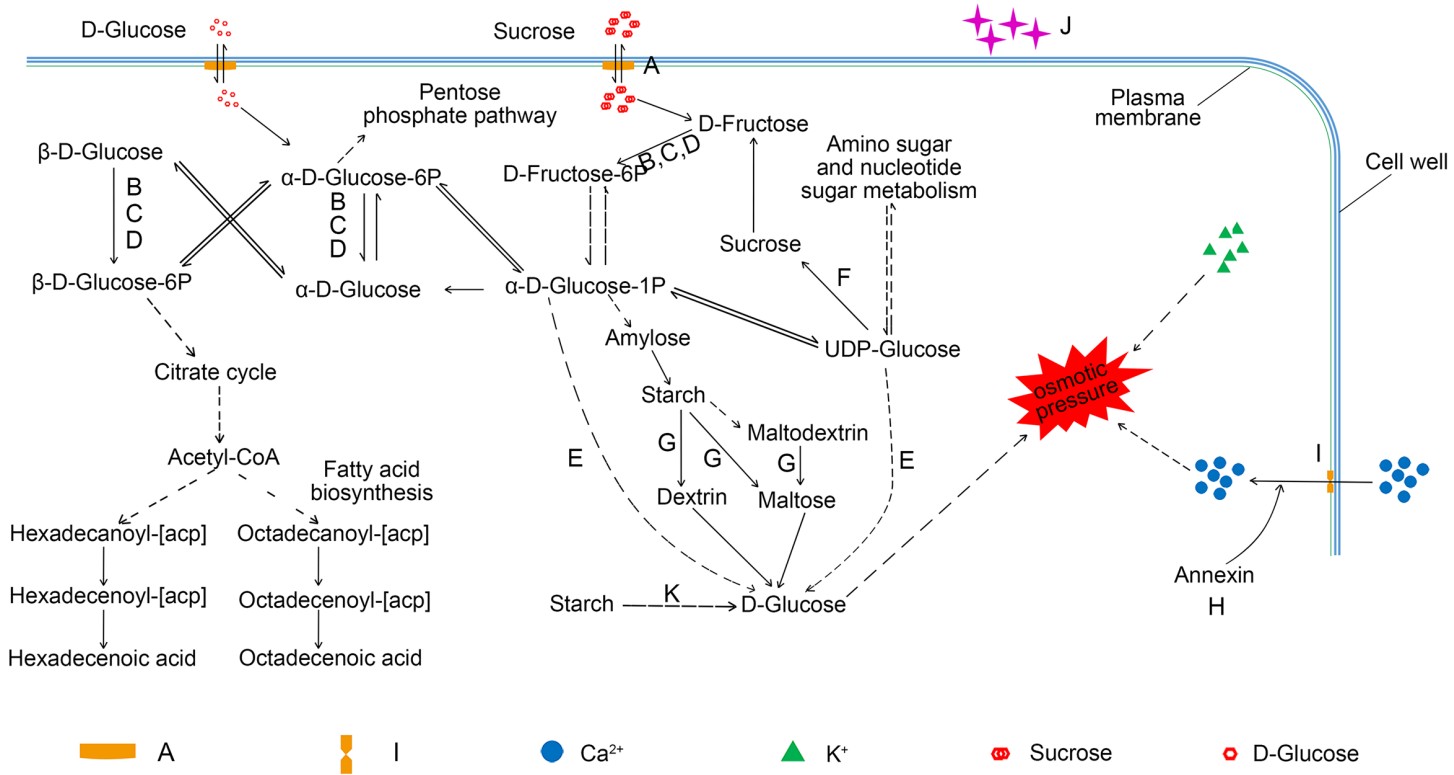

**Figure 6 An overview of the pathway for proteome metabolic changes between YM18 and *ZK001*.** (A) A0A1D6CCI3 (bidirectional sugar transporter SWEET); (B) A0A1D5SYC3 (cellular glucose homeostasis-related proteins); (C) A0A1D5RVB4 (cellular glucose homeostasis-related proteins); (D) A0A1D5VEI9 (cellular glucose homeostasis-related proteins); (E) A0A077S2F2 (beta-glucosidase activity-related protein); (F) W5B5R3 (sucrose synthase); (G) A0A1D5RR02 (beta-amylase); (H) A0A1D5RRV7 (annexin); (I) A0A1D5TN57 (calcium ion binding-related protein); (J) A0A1D5WGA3 (nutrient reservoir-related protein); (K) P93594 (beta-amylase).

hexokinase (A0A1D5SYC3, A0A1D5RVB4, and A0A1D5VEI9) after being broken down into D-fructose. D-fructose-6P converted to α-D-glucose-1P, which can be converted to D-glucose by β-glucosidase (A0A077S2F2) and synthesized into amylose. The starch formed from amylose can be broken down into D-glucose under the action of β-amylase (A0A1D5RR02 and P93594). UDP-glucose formed from amino sugar and nucleotide sugar can also be converted into D-glucose under the action of β-glucosidase (A0A077S2F2). The accumulation of D-glucose leads to a change in osmotic pressure in the lodicules. Additionally, α-D-glucose-1P can enter the pentose phosphate pathway and the fatty acid biosynthesis pathway through the formation of α-D-glucose-6P and β-D-glucose-6P. Soluble sugar can also enter and exit cells through the bidirectional sugar transporter. Furthermore, annexin (A0A1D5RRV7) can trigger calcium ion influx, increasing the osmotic pressure. Once the osmotic pressure changes, water accumulates in/is excreted from the cells of the lodicules and induces the expansion/shrinkage of the lodicules.

## CONCLUSIONS

The wheat mutant, *ZK001*, with its atrophied, thin and ineffective lodicules has lost the ability to push the lemma and palea apart in the flower development process. Compared

with YM18, *ZK001* showed a lower rate of *Fusarium* infection, presumably because of the cleistogamous phenotype. Furthemore, we speculate that the thin lodicule of *ZK001* results from its lower soluble sugar, calcium ion, and potassium ion contents, which are regulated by carbohydrate metabolic, protein transport, and calcium ion binding-related proteins. Though little is known about the mechanism of cleistogamy in wheat, we propose an overview of the metabolic pathway involving the carbohydrate that regulates the osmotic pressure of the lodicules. This study provides foundations for researchers to explore the mechanism of cleistogamy. Furthermore, it shows that it should be possible to generate cleistogamous wheat via conventional crossing, which would improve the FHB resistance of wheat and control the pollen-mediated gene flow of genetically modified wheat.

## ACKNOWLEDGEMENTS

We thank Prof. Xiue Wang, College of Agriculture, Nanjing Agricultural University, for providing FHB spore F0601. We thank Mr. Shiliang Li and Ms. Shengqun Zheng for field management. We also thank Mrs. Youwei Wu (graphic designer) for instruction in drawing Fig. 6.

### Funding

This work was funded by the Science and Technology Service program of the Chinese Academy of Sciences (KFJ-STS-ZDTP-002), the Key Program of the 13th five-year plan, Hefei Institutes of Physical Science, Chinese Academy of Sciences (CASHIPS) (No. kp-2017-21), and the Major Special Project of Anhui Province (16030701103). This study is also supported by the Natural Science Foundation of Anhui Provincial (1408085QC64), the Opening Fund of State Key Laboratory of Crop Genetics and Germplasm Enhancement (ZW2013003). The funders had no role in study design, data collection and analysis, decision to publish, or preparation of the manuscript.

### Grant Disclosures

The following grant information was disclosed by the authors:
Science and Technology Service program of Chinese Academy of Sciences:
KFJ-STS-ZDTP-002.
Hefei Institutes of Physical Science, Chinese Academy of Sciences (CASHIPS):
kp-2017-21.
Major special project of Anhui Province: 16030701103.
Natural Science Foundation of Anhui Provincial: 1408085QC64.
Opening Fund of State Key Laboratory of Crop Genetics and Germplasm Enhancement:
ZW2013003.

### Competing Interests

The authors declare that they have no competing interests.

## Author Contributions

- Caiguo Tang conceived and designed the experiments, performed the experiments, analyzed the data, prepared figures and/or tables, authored or reviewed drafts of the paper, approved the final draft.
- Huilan Zhang performed the experiments, contributed reagents/materials/analysis tools, prepared figures and/or tables.
- Pingping Zhang performed the experiments, contributed reagents/materials/analysis tools.
- Yuhan Ma analyzed the data, contributed reagents/materials/analysis tools, prepared figures and/or tables, authored or reviewed drafts of the paper.
- Minghui Cao performed the experiments, contributed reagents/materials/analysis tools.
- Hao Hu performed the experiments, analyzed the data.
- Faheem Afzal Shah contributed reagents/materials/analysis tools.
- Weiwei Zhao contributed reagents/materials/analysis tools.
- Minghao Li conceived and designed the experiments, performed the experiments, analyzed the data, prepared figures and/or tables, authored or reviewed drafts of the paper.
- Lifang Wu conceived and designed the experiments, authored or reviewed drafts of the paper.

## Data Availability

The mass spectrometry proteomics data is available at the PRIDE Archive: https://www.ebi.ac.uk/pride/archive/projects/PXD010188.

## Supplemental Information

Supplemental information for this article can be found online at http://dx.doi.org/10.7717/peerj.7104#supplemental-information.

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
