# Peer review of "iTRAQ-based quantitative proteome analysis reveals metabolic changes between a cleistogamous wheat mutant and its wild-type wheat counterpart"

_PeerJ, doi:10.7717/peerj.7104_

## Round 0.1 · original submission · Major Revisions

Please pay specific attention to the comments of Reviewer 2 in particular. The manuscript needs considerable scientific editing. In particular, it requires a professional English editor. The chemistry (naming of molecules) and other details is often erroneous. In particular, the graphs and tables have some revisions needed. The overstating of significance is a real problem. If statistics was not significant, than up or down regulation was also not significant and cannot be discussed in the manuscript as altered. Please revise accordingly and I look forward to an improved version in the next weeks.

·

Basic reporting

The article is generally well written and presented. I have corrected some English and typographical errors as I found them but they are minimal.
Referencing appears appropriate.

Experimental design

No comment

Validity of the findings

I enjoyed reading the manuscript and found it followed a logical progression leading to the conclusions.

Additional comments

When presenting chemical names it may be advisable in future to get a professional chemist to check your naming. You clearly have an issue with this in the manuscript that needs correcting. The Figure 6 is good though so please check that this has been produced by members of the team and does not require acknowledgement if produced by others.

Reviewer 2 ·

Basic reporting

While this paper presents the results of important findings that should be of use to plant breeders, the use of English language is deficient throughout the paper and will require extensive editing. In addition, there are numerous mistakes (e.g.FADs are referred to as FEDs in the diagrams and statistical results are not consistently interpreted correctly or are ignored). The structure of the paper is sound and the results are meaningful, but there are many errors that detract from the paper and make it unpublishable in its current form.

In the section "Formatting of Mathematical Components" (which is a totally inappropriate section heading, by the way), there seem to be some reporting errors. The authors state that there are differences between YM18 and ZK001 from YAS to AS stages, but the results suggest that the differences lay between WAS and AS. The extent of the difference cited by the authors is also questionable; they cite a difference of 39% for ZK001, but from Table 2 this difference appears to be 50% (and it's not clear where the "slight difference" for YM18 has been determined).

The formulae used for calculating starch and sugar levels are a bit difficult to follow because they extend beyond one line. More importantly, there seems to be something missing in the starch content equation: the last term ("of sample") is incomplete. The authors also misuse "respectively", using it in places where it is not needed (e.g. line 289).

The specifications of the HPLC columns is vague; the first dimension in each case (1.9 um and 3 um, respectively) presumably refers to the particle size, but this is not spelled out (it's just inserted with no punctuation or clarifying information). The authors have not used subscripts were needed in many places (e.g. C18 and H2O). The authors state that they slightly modified the extraction procedures for proteins published by Yang et al. (2013), but failed to say what those modifications were.

The figure captions are lacking in necessary detail. In the caption for Figure 2, there should be a distinction between what is illustrated in panels A & C vs B & D. Figure 6 presents a key below the figure that appears to be a heat map, but there's no indication what the bars represent (presumably the level of each of the proteins during the 4 flowering stages for YM18 and ZK001, but this is just a guess--the reader shouldn't have to guess!). The caption for Table 1 is also very deficient in details; presumably this table shows infection by Fusarium head blight by the four varieties, but this is not specified (the column heading should be Variety, not Species).

Experimental design

The experiments appear to have been well designed and adequately replicated. Most of the data have been appropriately analyzed, but it appears that the starch and sugar levels during the YAS and AS stages were not analyzed (they may have been analyzed, but no statistical significance is mentioned in the results on this section, lines 279-284 and Fig. 2).

Validity of the findings

Data have not been properly interpreted in the section Accumulation patterns of DAPs and verification of DAPs of interest (note that I have reworded this title to be grammatically correct). From lines 358-374, there are numerous instances of statements that are not true because statistical significance has apparently been overlooked. For example, the authors state that the A0A1D5RR02 gene was downregulated during WAS and GAS for ZK001, but these differences are not significant (as indicated by the letters over the bars in Fig. 5C). There are many similar misstatements in the following sentences. The sentence regarding A0A1D5SYC3 and A0A1D5VE19 on lines 363-366 also makes no sense.

Additional comments

There seems to be a solid body of work that was conducted, but careless reporting and poor English make this paper look quite deficient. I think the quality of the work was good, but the presentation is quite deficient. I have not offered comments on all of the English grammar problems because they are too extensive.

---

## Round 0.2 · Minor Revisions

The reviewer, and I have reviewed the revised manuscript and find it requires several minor revisions. It is however improved over last version.

Reviewer 2 ·

Basic reporting

.

Experimental design

.

Validity of the findings

.

Additional comments

The manuscript is much improved, but some additional minor editing is still required.
See attachment.

Annotated reviews are not available for download in order to protect the identity of reviewers who chose to remain anonymous.

---

## Round 0.3 · accepted · Accept

Thanks for addressing the comments of all reviewers.